# miRNA and Its Implications in the Treatment Resistance in Breast Cancer—Narrative Review of What Do We Know So Far

**DOI:** 10.3390/ncrna11060078

**Published:** 2025-11-18

**Authors:** Isabela Anda Komporaly, Adelina Silvana Gheorghe, Lidia Anca Kajanto, Elena Adriana Iovănescu, Bogdan Georgescu, Raluca Ioana Mihăilă, Andreea Mihaela Radu, Daniela Luminița Zob, Mara Mădălina Mihai, Mihai Teodor Georgescu, Dana Lucia Stănculeanu

**Affiliations:** 1Department of Oncology, “Carol Davila” University of Medicine and Pharmacy, 020021 Bucharest, Romania; isabelakomporaly@gmail.com (I.A.K.); dumitrescu.elena.adriana@gmail.com (E.A.I.); dr_bogdan.georgescu@icloud.com (B.G.); dr.ralucamihaila@yahoo.com (R.I.M.); stoicaandreea06@gmail.com (A.M.R.); mihai.georgescu@umfcd.ro (M.T.G.); dlstanculeanu@gmail.com (D.L.S.); 2Memorial Hospital, 013812 Bucharest, Romania; 3Department of Medical Oncology I, “Prof. Dr. Alexandru Trestioreanu”, Institute of Oncology, 022328 Bucharest, Romania; 4Neolife Hospital, 077190 Bucharest, Romania; 5MedEuropa Clinic, 022343 Bucharest, Romania; 6Department of Medical Oncology II, “Prof. Dr. Alexandru Trestioreanu”, Institute of Oncology, 022328 Bucharest, Romania; danielazob@yahoo.com; 7Department of Oncologic Dermatology, “Carol Davila” University of Medicine and Pharmacy, 020021 Bucharest, Romania; mara.mihai@umfcd.ro; 8Elias Emergency University Hospital, 011461 Bucharest, Romania; 9Department of Radiotherapy II, “Prof. Dr. Alexandru Trestioreanu”, Institute of Oncology, 022328 Bucharest, Romania

**Keywords:** breast cancer, microRNAs (miRNAs), non-coding RNA biomarkers, treatment resistance, tumor heterogeneity, molecular subtypes, biomarkers, precision oncology

## Abstract

Breast cancer remains a leading cause of cancer-related mortality worldwide, with treatment resistance and tumor heterogeneity posing major clinical challenges. MicroRNAs (miRNAs), small non-coding RNAs regulating gene expression, have emerged as key players in breast cancer biology, influencing tumor initiation, progression, and therapy resistance. This narrative review synthesizes recent evidence on the involvement of miRNAs in breast cancer subtypes and their impact on treatment response. Notably, miR-155, miR-503, and miR-21 have shown potential as non-invasive biomarkers and modulators of pathways such as PI3K-Akt, MAPK, and TNF signaling. Additionally, exosomal miRNAs may reflect chemoresistance profiles and predict pathological response to neoadjuvant therapy. Emerging data also support the use of specific miRNAs to sensitize tumors to radiotherapy or modulate immune checkpoints like PD-L1 in triple-negative breast cancer. However, challenges persist regarding standardization, sample types, and study heterogeneity. Further translational research is needed to validate miRNA signatures and their utility in guiding personalized treatment. By highlighting mechanistic insights and potential clinical applications, this review aims to contribute to the ongoing efforts of integrating miRNAs into precision oncology for breast cancer.

## 1. Introduction

Breast cancer is the most commonly diagnosed cancer and a primary cause of cancer-related mortality among women globally. Cancer Statistics 2024 estimates 313,510 new breast cancer cases and 42,250 deaths in the United States for 2024, highlighting the ongoing worldwide burden and clinical significance of this illness [1]. Breast cancer treatment includes surgical methods, radiotherapy, conventional chemotherapy, targeted therapy, and, more recently, immunotherapy. However, despite technological advancements, some patients exhibit suboptimal responses to treatment. Tumor heterogeneity, treatment resistance, and recurrence are key factors in determining long-term survival [2,3,4].

Evidence suggests that certain molecular subtypes of breast cancer respond better than others to treatment. For example, triple-negative breast cancer is a molecular subtype that does not benefit from hormonal or anti-HER therapies, relying primarily on chemotherapy, and more recently, on immunotherapy or targeted therapies such as PARP inhibitors. These are effective not only in BRCA1/2-mutated tumors but also in cases with other homologous recombination repair (HRR) deficiencies or genomic instability characterized by high HRD scores [5,6]. Although the initial response is usually favorable, recurrence is frequently observed. Alongside molecular heterogeneity, treatment resistance, and recurrence are factors that impact survival [7].

Numerous predictive factors in breast cancer have been studied, and one emerging area of research involves microRNAs (miRNAs) [2]. miRNAs are families of small, endogenous RNA molecules, 20–24 nucleotides long, first identified in 1993 [8]. These molecules are involved in regulating multiple biological phenomena across numerous species, primarily through post-translational modifications [3,4]. The biological processes regulated by miRNAs, as noted in scientific literature, include cellular homeostasis, proliferation, differentiation, and apoptosis [7,9,10]. Approximately 60% of the human genome is regulated by miRNAs, with one gene potentially regulated by multiple miRNAs, and one miRNA regulating multiple genes [2,3].

Aberrant expression of miRNAs interferes with key processes in cancer progression, including tissue invasion, metastasis, resistance to therapy, and inhibition of apoptosis [11,12]. Recent research highlights the role of miRNAs in tumor initiation and development by influencing cancer stem cell properties, such as self-renewal capacity and drug resistance [13,14,15]. Notably, miRNAs have the ability to regulate both gene expression and the function of their associated products within crucial signaling pathways, including Wnt/β-catenin, Notch, and others, which are essential for the maintenance, proliferation, and activity of tumor stem cells [16,17,18,19].

Preclinical studies suggest that therapeutic modulation of miRNAs, achieved by inhibiting oncogenic miRNAs and replacing deficient tumor suppressor miRNAs, could improve cancer treatment.

This scientific paper aims to synthesize the current understanding of the relationship between specific miRNA families, breast cancer molecular subtypes, and treatment resistance by analyzing high-quality systematic reviews and meta-analyses published in specialized journals. This approach was chosen to ensure the inclusion of clinically validated findings and to provide a comprehensive overview based on aggregated evidence, rather than isolated experimental data. To complement this approach, additional narrative integration of preclinical and translational studies was performed to highlight key mechanistic insights and emerging clinical implications of miRNAs in therapy resistance. The literature selection was guided by scientific relevance rather than exhaustive database retrieval, emphasizing recent peer-reviewed publications that elucidate molecular pathways, predictive biomarkers, and therapeutic potential. By combining evidence from both aggregated analyses and representative experimental reports, this review provides a balanced synthesis of current knowledge and identifies critical gaps that warrant further investigation in the field of precision oncology.

## 2. Breast Cancer and miRNA

### 2.1. miRNA Dysregulation in Breast Cancer, Functional Pathways and Mechanisms

#### 2.1.1. Diagnostic and Prognostic Relevance of miRNA Dysregulation

A 2021 meta-analysis by Thu N.N.H. and colleagues, published in Molecular Biology, investigated miR-16 expression patterns in breast cancer. The research compiled data from several sources to ascertain the standardized mean difference (SMD), subsequently doing Chi-square analysis, which indicated significant disparities in miR-16 expression between breast cancer patients and healthy persons. The observed negative SMD of −0.56, accompanied by a Chi-square value of 62.62 (*p* = 0.05), indicated miR-16 dysregulation in breast cancer. The significant fluctuation in stability values (SV) of miR-16 expression further underscored its unreliability as a regulatory gene in this setting [20].

Another 2021 meta-analysis in Breast Cancer by Xuemin Liu and colleagues demonstrated the importance of miR-155 as an early diagnostic marker for breast cancer. miR-155 is considered one of the most promising members of the miRNA family, examined as a potential biomarker for cancer detection and survival prognosis. Nineteen published publications suggest that miR-155 holds considerable potential as a novel non-invasive biomarker for breast cancer screening. Moreover, integrated bioinformatic analysis identified numerous crucial hub genes and pathways that may clarify the remarkable biomarker characteristics of miR-155 and provide insights into the specific etiology and mechanisms associated with the initiation and progression of breast cancer. However, sample sizes in most studies are rather small, and the diagnostic relevance of miR-155 expression in breast cancer remains unclear. This meta-analysis meticulously assessed clinical trials about the applications of miR-155 in breast cancer diagnosis in recent years. The meta-analysis aimed to clarify the clinical significance of miR-155 expression in breast cancer. Research demonstrates that miR-155 displays more precision and improved diagnostic characteristics in breast cancer detection compared to traditional methods [21].

#### 2.1.2. Functional Pathways Influenced by Dysregulated miRNAs

Extensive study is currently focused on examining the functional significance of miR-155 in cancer development and malignant progression. Nonetheless, the essential molecular pathways behind cancer proliferation remain predominantly unidentified. Gene Ontology (GO) study has associated miR-155 with critical biological processes, fundamental cellular components, and notable molecular interactions. Furthermore, KEGG functional enrichment analysis has shown numerous critical pathways intimately associated with the onset and advancement of breast cancer. The TNF signaling system is pivotal in sustaining homeostasis and promoting disease progression [22].

The prolactin pathway, a hormone crucial for normal breast development and lactation, has been linked to the advancement of breast cancer [23].

T-cell receptor signaling is crucial in the host adaptive immune system, and T-cell-based adoptive immunotherapy has proven effective for multiple cancer types. Expression of deviant T-cell receptors may result in carcinogenesis [24].

HIF-1 is acknowledged as a vital element in altering tumors’ transcriptional responses to hypoxia and is integral to key facets of cancer biology, such as angiogenesis, cell survival, glucose metabolism, and invasion [25].

The PI3K-Akt signaling system, often hyperactivated in breast carcinomas, is crucial for regulating biological processes including cell growth, differentiation, migration, survival, angiogenesis, and metabolism. Prior research has underscored the essential role of miRNAs in cellular processes associated with tumor proliferation, migration, and breast cancer metastasis through the modulation of the oncogenic PI3K/Akt pathway. The insights derived from these enrichment studies may enhance the knowledge of miR-155′s function in the initiation and progression of breast cancer [26,27].

Protein–protein interaction (PPI) analysis is acknowledged as a foundational approach for elucidating associations among various genome-wide proteins, providing novel insights into the interpretation of protein function. The PPI network was established using miR-155 target genes, and the 10 principal hub genes were discovered. The hub genes were mostly linked to numerous significant pathways, the majority of which were validated as associated with breast cancer. Moreover, research indicates that proteoglycans can modulate both normal and pathological processes, including morphogenesis, tissue repair, inflammation, vascularization, and cancer metastasis [26,27].

Recent findings indicate that the Jak-STAT signaling system governs nearly all immune regulatory activities, encompassing tumor cell identification and tumor-induced immune clearance. Alterations in this system may facilitate breast cancer progression and metastasis, whereas inhibiting JAK-STAT signaling in breast cancer could provide viable therapeutic approaches to address treatment resistance [28,29].

The MAPK signaling pathway is essential for orchestrating a complex network that governs various physiological processes, including cell proliferation, differentiation, and apoptosis. The dysregulation of the MAPK signaling pathway has been associated with the development of several human malignancies, including breast cancer. Hub genes may function as potential prognostic indicators and therapeutic targets for individuals with breast cancer [30,31].

The examination of the PPI network modules identified two principal modules for KEGG pathway enrichment, indicating that the related genes were primarily associated with essential biological pathways. The cell cycle is a crucial regulator of cancer development, directly influencing cell proliferation and tumor formation, with its dysregulation propelling cancer progression [32]. Moreover, focal adhesion kinase (FAK), a cytoplasmic tyrosine kinase, is integral to various cancer-associated processes, such as tumor invasion, metastasis, epithelial-mesenchymal transition (EMT), and the maintenance of cancer stem cells. Sphingolipids, a significant category of bioactive lipids, play crucial roles in fundamental cellular processes like proliferation and death, and they influence multiple facets of cancer progression [32,33].

PPI study revealed probable mechanisms by which miR-155 contributes to the onset and progression of breast cancer. Nonetheless, further research is necessary to authenticate hub genes and pathways, and to uncover more profound mechanisms. This review has numerous notable limitations that must be acknowledged. Firstly, the lack of standardized miR-155 cutoff values across the included diagnostic tests may have contributed to the observed heterogeneity [34]. Further studies are required to establish an optimal cutoff value that ensures greater diagnostic accuracy. Secondly, the sample sizes for plasma, whole blood, tissue, and urine were relatively small, making it difficult to draw definitive conclusions regarding the reliability of miR-155 as a diagnostic marker for breast cancer from these sources. Additional research is necessary to assess the potential of miR-155 across various sample types [35]. Thirdly, the limited number of studies evaluating miR-155 diagnostic performance across different ethnic groups complicates its clinical applicability. Further investigation is required to ascertain its diagnostic precision among varied populations. Moreover, the included studies aggregated data from patients across diverse TNM stages (I–IV), resulting in considerable heterogeneity in breast cancer diagnosis [35]. The absence of patient-level data complicates the assessment of miR-155 as a dependable biomarker for early breast cancer detection. Moreover, tumors characterized by BRCA1 deficiency demonstrate elevated levels of miR-155, which may be pertinent for identifying potential responders to PARP-1 inhibitors [34,35]. Ultimately, the absence of experimental validation necessitates additional investigation of miR-155′s functions and mechanisms through both in vivo and in vitro studies.

#### 2.1.3. Non-Coding RNAs in the Regulation of Epithelial–Mesenchymal Transition in Breast Cancer

##### Overview of EMT in Breast Cancer

Epithelial–mesenchymal transition (EMT) is a reversible biological process enabling epithelial cells to acquire mesenchymal traits, including enhanced migratory capacity, invasiveness, and resistance to apoptosis. Epithelial-mesenchymal transition (EMT) is crucial in breast cancer for tumor progression, metastatic dissemination, and the development of resistance to many therapies, particularly in aggressive forms such as triple-negative breast cancer (TNBC) [36,37]. Epithelial-mesenchymal transition (EMT) is molecularly regulated by transcription factors such as Snail (SNAI1), Slug (SNAI2), ZEB1, ZEB2, TWIST1, and TWIST2, which inhibit epithelial markers like CDH1 (encoding E-cadherin) and enhance mesenchymal genes such as VIM (vimentin) and FN1 (fibronectin) [38,39].

The initiation and sustenance of epithelial-mesenchymal transition (EMT) encompass both transcriptional regulation and intricate signaling networks, including as TGF-β, Wnt/β-catenin, PI3K/AKT, and MAPK pathways, alongside comprehensive epigenetic reprogramming. In recent years, non-coding RNAs (ncRNAs), especially microRNAs (miRNAs) and long non-coding RNAs (lncRNAs), have emerged as essential regulators of epithelial-mesenchymal transition (EMT), functioning either upstream to initiate the process, downstream to maintain it, or within feedback loops to precisely adjust the epithelial-mesenchymal equilibrium [40,41].

##### EMT-Suppressive miRNAs

A large body of evidence supports the role of specific miRNAs as potent suppressors of EMT in breast cancer. Among them, the miR-200 family (miR-200a, miR-200b, miR-200c, miR-141, and miR-429) is one of the most extensively studied. These miRNAs directly target ZEB1 and ZEB2, leading to the restoration of E-cadherin expression, stabilization of epithelial characteristics, and reduction in invasive and metastatic capabilities [42,43]. Loss of miR-200 expression is frequently observed in basal-like breast tumors and is associated with distant metastasis and poor clinical outcomes.

Another well-characterized EMT-suppressive miRNA is miR-34a, a direct transcriptional target of p53. miR-34a inhibits SNAI1 and SNAI2, thereby repressing EMT transcription factors, reducing cancer stem-like features, and sensitizing TNBC cells to chemotherapy [44]. miR-30a exerts its anti-EMT function by targeting SOX4 within the TGF-β/SMAD pathway, resulting in decreased N-cadherin and increased E-cadherin expression [45]. miR-205-5p negatively regulates ZEB1 and ZEB2, and its expression inversely correlates with the presence of mesenchymal-type circulating tumor cells in metastatic breast cancer [46]. miR-145 contributes to EMT suppression by targeting FSCN1, an actin-bundling protein essential for cell motility [47], while miR-29b modulates TET1-mediated demethylation of ZEB2, thereby limiting the mesenchymal shift [48].

These miRNAs act as molecular brakes on EMT, and their downregulation facilitates the transition toward a mesenchymal phenotype, contributing to tumor aggressiveness and treatment resistance.

##### EMT-Promoting miRNAs

In contrast, a distinct group of miRNAs functions as oncogenic drivers, or “oncomiRs,” by promoting EMT and enhancing metastatic potential. miR-9 is one such regulator, directly targeting CDH1 and disrupting cell–cell adhesion, thereby promoting invasion [49]. The miR-221/222 cluster is strongly associated with EMT induction, particularly in aggressive luminal B tumors, where it suppresses epithelial markers and upregulates mesenchymal gene expression [50].

miR-10b, transcriptionally activated by TWIST1, downregulates HOXD10, leading to the activation of pro-invasive genes such as RHOC and facilitating migration and invasion [51]. Elevated circulating miR-10b levels have been proposed as a marker of metastatic potential in breast cancer patients. miR-181a promotes EMT through activation of TGF-β signaling, which not only enhances invasion but also supports the acquisition of stem-like features [52].

These pro-EMT miRNAs contribute to metastatic dissemination and frequently confer resistance to chemotherapeutic and endocrine treatments, highlighting their importance as therapeutic targets.

##### lncRNAs as Master Regulators of EMT

Long non-coding RNAs have emerged as essential regulators of epithelial-mesenchymal transition (EMT) in breast cancer, functioning through many mechanisms including transcriptional regulation, chromatin remodeling, and miRNA sponging. HOTAIR is one of the most prominent pro-EMT lncRNAs, known to recruit Polycomb Repressive Complex 2 (PRC2) and LSD1 to epithelial gene loci, resulting in chromatin modifications that silence epithelial traits and promote metastasis [53]. MALAT1 enhances EMT by sequestering tumor-suppressive miRNAs, including miR-1 and miR-200c, and by modulating alternative splicing of EMT-related transcripts [54].

Antisense lncRNAs such as ZEB1-AS1 and ZEB2-AS1 stabilize their respective mRNAs and recruit histone methyltransferases, such as MLL1, to promote transcriptional activation of ZEB family members [55,56]. NEAT1 has been linked to both EMT induction and therapy resistance; its silencing restores epithelial characteristics and increases drug sensitivity [57]. TUG1 promotes invasion by interacting with p53-related pathways and by sponging EMT-suppressive miRNAs [58].

These lncRNAs function as molecular scaffolds and decoys, exerting a multi-layered control over EMT and significantly influencing the metastatic trajectory of breast tumors.

##### Emerging lncRNAs and ceRNA Networks

Recent studies have identified novel lncRNAs, including SNHG16 and SNHG3, which are capable of promoting EMT in breast cancer. These lncRNAs can be packaged into tumor-derived exosomes, facilitating intercellular communication and enhancing metastatic potential [59,60]. Knockdown experiments demonstrate that suppression of these transcripts leads to reversal of EMT, reduced migratory capacity, and impaired metastasis formation in vivo.

Many pro-EMT lncRNAs operate within competing endogenous RNA (ceRNA) networks, wherein they act as molecular sponges for EMT-suppressive miRNAs such as the miR-200 family and miR-34a. By sequestering these miRNAs, lncRNAs de-repress EMT transcription factors, including ZEB1, SNAI1, and TWIST1, amplifying the transcriptional reprogramming toward a mesenchymal phenotype. This convergence on oncogenic pathways such as TGF-β/SMAD, PI3K/AKT, and MAPK signaling underscores the potential of targeting multiple ncRNAs simultaneously as a therapeutic strategy.

##### Therapeutic Implications

The regulatory interplay between ncRNAs and EMT presents multiple opportunities for therapeutic intervention. Restoration of EMT-suppressive miRNAs through synthetic mimics, inhibition of pro-EMT miRNAs using antagomiRs or locked nucleic acid (LNA) inhibitors, and silencing of pro-EMT lncRNAs with siRNAs or antisense oligonucleotides are among the approaches under investigation [61,62]. Given the redundancy and complexity of EMT regulation, combination strategies targeting both miRNAs and lncRNAs, along with their associated signaling pathways, may yield synergistic effects in preventing metastasis and overcoming drug resistance in breast cancer.

### 2.2. miRNAs and Chemoresistance

A noteworthy area in breast cancer research is the link to chemoresistance, which impacts survival. Despite technological advances and an improved understanding of carcinogenesis mechanisms, predicting which breast cancer patients will respond to treatment remains challenging. The lack of predictive capacity often leads to multiple therapy lines to find an option that yields a therapeutic response. Does miRNA have predictive value? A 2021 meta-analysis published in Cancers by America Campos and collaborators [63] focused on studying the presence and possible chemoresistant or chemosensitive role of circulating or exosome-derived miRNAs in breast cancer patients undergoing neoadjuvant chemotherapy, analyzing multiple studies, four of which focused on breast cancer [64,65,66].

Stevic and colleagues examined 384 distinct miRNAs using RT-PCR in exosomes derived from a cohort of breast cancer patients (*n* = 15) who had chemotherapy, with or without complete pathological response (pCR) [67]. A total of 45 exosomal miRNAs were found to be significantly dysregulated across the analyzed samples. To further investigate exosomal miRNA expression, a larger cohort of breast cancer patients (*n* = 435) was assessed using plasma samples collected before neoadjuvant chemotherapy. These results were compared with those from a control group of healthy individuals (*n* = 20) and a smaller subset of breast cancer patients (*n* = 9) who had undergone neoadjuvant chemotherapy prior to surgery. Notably, miR-155, miR-27a, miR-376c, and miR-376a showed reduced levels in treated patients compared to their pre-treatment samples. The expression levels of these miRNAs in post-treatment patients closely mirrored those found in healthy controls. The researchers proposed that these miRNAs may be disseminated from the main tumor into the bloodstream, acting as possible biomarkers of disease state in previously treated patients. Additionally, their findings demonstrated that in both univariate and multivariate models, miR-155 and miR-301 were significant predictors of pCR, suggesting a better response to carboplatin-based therapy in these cases. However, the authors acknowledged that the limited number of patients who received neoadjuvant chemotherapy made it challenging to draw definitive statistical conclusions about the impact of treatment on the specified miRNA expression levels [67].

A collection of exosomal miRNAs, comprising miR-105, miR-21, miR-222, and miR-221, derived from serum samples of breast cancer patients (*n* = 47) prior to and during neoadjuvant chemotherapy, has been associated with an indirect influence on chemotherapy response. Rodriguez-Martínez’s study revealed a significant correlation between exosomal miR-21 levels and tumor size (*p* = 0.039), with diminished expression of this miRNA noted in 37 of 47 patients receiving neoadjuvant chemotherapy. Conversely, untreated patients displayed exosomal miR-222 in conjunction with proliferation markers, including Ki67 (*p* = 0.05). The study indicated that breast cancer patients with affected lymph nodes who underwent neoadjuvant chemotherapy exhibited significantly reduced levels of exosomal miR-221 compared to those without lymph node involvement receiving the same treatment. Additionally, increased concentrations of exosomal miR-105 and miR-21 were observed in a group of six patients with metastatic disease in contrast to eight healthy individuals. The researchers sought to assess the viability of exosomal miRNAs as adjunct clinical instruments to enhance breast cancer diagnosis and prognosis. The researchers determined that the detection of exosomal miR-21, miR-105, miR-221, and/or miR-222 may facilitate the prediction of early relapse and function as biomarkers for recognizing initial resistance to doxorubicin/cyclophosphamide treatment [64].

Zhong and colleagues identified 22 miRNAs that were upregulated in exosomes from breast cancer cell lines resistant to epirubicin, docetaxel, or vinorelbine, in comparison to exosomes from their respective parental cell lines. The researchers conducted a detailed analysis of a specific subset of miRNAs in biopsy specimens obtained prior to neoadjuvant chemotherapy and in surgically excised tumor samples from 23 breast cancer patients. Their results indicated that 12 of the 22 chosen miRNAs were significantly elevated after neoadjuvant treatment. Significantly, only miR-574-3p exhibited a statistically meaningful correlation with disease progression. Its expression was specifically modulated in tissue samples from patients with stable or progressing disease, unlike those who demonstrated a partial response to chemotherapy (*p* = 0.027) [68].

Bovy and colleagues presented compelling in vitro evidence indicating that exosomal miR-503 from endothelial cells may function as a tumor suppressor in breast cancer. Their research indicated that this exosomal miRNA diminished the proliferative and invasive capabilities of receptor-positive breast cancer MDA-MB-231 cells [68,69]. Furthermore, they noted that endothelial cells exposed to epirubicin and paclitaxel displayed elevated levels of exosomal miR-503 in comparison to untreated endothelial cells. A prospective research including breast cancer patients receiving neoadjuvant chemotherapy (*n* = 17) revealed increased circulating levels of miR-503 compared to those who received surgery alone (*n* = 12). Bovy et al. did not directly isolate exosomes from the plasma of breast cancer patients to investigate the effect of miR-503; nonetheless, their findings imply possible modifications in the tumor microenvironment due to treatment. Recent research reveal that miR-503 can directly bind to and inhibit BLACAT1 (bladder cancer-associated transcript 1), a long non-coding RNA associated with developing chemotherapy resistance in T47D and MCF7 breast cancer cells [70]. Additionally, miR-503 is recognized as a constituent of the miRNA cluster miR-424(322)/503, whose reduction in breast cancer patients is associated with the development of aggressive breast cancer and has been linked to the enhancement in in vivo paclitaxel resistance via the upregulation of two of its targets: B-cell lymphoma 2 (BCL-2) and insulin-like growth factor-1 receptor (IGF1R) [71].

### 2.3. miRNA and PD-L1/PD-1 Immune Checkpoint Inhibition

The PD-L1/PD-1 immune checkpoint pathway is well-recognized in triple-negative breast cancer (TNBC). Emerging data suggests that tumoral PD-L1 may play a role in the progression of TNBC. Although conventional immune checkpoint inhibitors have improved the prognosis for TNBC patients, their effects mostly focus on enhancing anticancer immune responses rather than significantly altering oncogenic signaling pathways within tumor cells. Furthermore, conventional immune checkpoint inhibitors are unable to suppress the de novo synthesis of oncoproteins, such as PD-L1, in tumor cells. Emerging evidence indicates that the reestablishment of specific miRNAs can suppress tumoral PD-L1 and hinder the advancement of TNBC. Considering that miRNAs can target many mRNAs, miRNA-based gene therapy may provide an effective approach to inhibit de novo oncoprotein synthesis, such as PD-L1, restore anticancer immune responses, and regulate several intracellular signaling pathways in TNBC.

In 2021, Mahdi Abdoli Shadbad and associates published an extensive review in Genes on the impact of miRNA-mediated PD-L1 suppression in triple-negative breast cancer [72].

Recent studies have discovered a subset of miRNAs that can convert the immunosuppressive tumor microenvironment into a pro-inflammatory state by downregulating PD-L1 expression and modifying oncogenic signaling pathways in TNBC cells. This extensive review emphasizes PD-L1-inhibiting miRNAs and investigates their therapeutic potential in curtailing TNBC progression [72]. Furthermore, it addresses prospective ways for enhancing the selective delivery and targeted administration of PD-L1-inhibiting miRNAs. The systematic review indicated that reinstating the expression of miR-200c-3p, miR-424-5p, miR-138-5p, miR-34a-5p, miR-570-3p, miR-383-5p, miR-3609, miR-195-5p, and miR-497-5p can diminish PD-L1 expression in tumors, alter the tumor microenvironment to a pro-inflammatory condition, decrease tumor proliferation and migration, enhance chemosensitivity in tumor cells, promote apoptosis, induce cell cycle arrest, inhibit clonogenic capacity, and modulate critical oncogenic signaling pathways in TNBC cells [72].

### 2.4. Radiotherapy and miRNAs

Radiotherapy (RT) is commonly used as an adjuvant treatment for breast cancer and is currently being explored for its potential to enhance neoadjuvant therapies. However, radioresistance remains a significant challenge, highlighting the need for novel biomarkers to identify patients who may benefit from RT integration in breast cancer treatment. MicroRNAs (miRNAs) are key regulators of gene expression that influence cancer cell responses to ionizing radiation (IR) and can be assessed through tumor tissue or liquid biopsy. In 2022, Nhu Hanh To and colleagues published a systematic review in Breast Cancer Research and Treatment, evaluating the association between miRNAs and radiation response in triple-negative breast cancer (TNBC), along with their predictive and prognostic significance [73].

Thirty-five microRNAs (miRs) were discovered as associated with triple-negative breast cancer (TNBC), comprising 21 that were downregulated, 13 that were upregulated, and 2 that demonstrated dual expression in this malignancy. The modulation of the expression of certain miRs enhances radiosensitivity, particularly miR-7, -27a, -34a, -122, and let-7, which have been extensively researched, albeit solely in experimental models. The microRNAs most affected by triple-negative breast cancer (TNBC) response to ionizing radiation (IR) include miR-7, -155, -27a, -211, -205, and -221, whereas miR-21, -139-5p, -33a, and -210 correlate with patient outcomes in TNBC following radiotherapy. In conclusion, microRNAs are emerging biomarkers and radiosensitizers in triple-negative breast cancer, warranting more investigation. The dynamic assessment of circulating microRNAs may enhance the monitoring and efficacy of radiation in triple-negative breast cancer [73].

### 2.5. miRNAs in Her2neu-Enriched Breast Cancer

A particular biological subtype of breast cancer is the Her2neu-enriched variant. Recent discoveries about the HER2/neu signaling pathway have facilitated the development of personalized therapeutic agents. Monoclonal antibodies targeting the HER2/neu receptor and an oral tyrosine kinase inhibitor that reversibly affects HER1, HER2, and the epidermal growth factor receptor are presently utilized with traditional chemotherapeutic drugs to improve clinical and oncological outcomes [74,75]. The amalgamation of Trastuzumab, Carboplatin, and Docetaxel has demonstrated exceptional survival rates in HER2/neu-positive tumor cohorts [76]. The sensitivity of tumors to neoadjuvant chemotherapy is a crucial prognostic factor in breast cancer, with biomarkers such as pCR—defined by the absence of residual or invasive disease following neoadjuvant treatment—associating with positive predictive value for survival outcomes. Patients achieving pathological complete response demonstrate improved long-term survival [77,78,79,80].

Despite reports of pCR rates reaching 70% in HER2/neu-enriched disease, current translational research efforts have focused on identifying novel, reliable, and sensitive biomarkers that can rival established clinicopathological markers (e.g., estrogen [ER], progesterone [PgR], HER2/neu receptor status, Ki-67 expression profiles, multigene expression tests) in predicting oncological outcomes, including responses to neoadjuvant therapies. However, a lack of effective biomarkers remains for evaluating treatment outcomes in HER2/neu-overexpressing breast cancer [81,82].

In 2022, Davey MG and associates published a systematic review in Breast Cancer (Auckl) regarding the involvement of miRNA in forecasting responses to neoadjuvant therapy in HER2/neu-enriched molecular subtype breast cancer [83].

Evidence indicates that a panel of miRNAs may aid in stratifying patients with an increased probability of responding to neoadjuvant therapy [84,85]. The present research finds 73 miRNAs potentially relevant for predicting responses to traditional neoadjuvant treatment regimens, including 41 miRNAs indicative of a heightened likelihood of reaching pCR.

The clinical relevance of miRNA signatures lies in their potential to enhance patient prognosis, predict therapy response, and refine existing treatment strategies. This systematic review is the first to consolidate previous foundational research to identify miRNAs with expression profiles capable of predicting response or pCR to neoadjuvant therapies. As part of the translational research conducted in the NeoALTTO study, pCR assessment identified ct-miR-140a-5p, ct-miR-148a-3p, and ct-miR-374a-5p as predictive biomarkers in patients receiving Trastuzumab, with a total predictive accuracy of 54%, compared to 0% in patients with high versus low expression levels. Additionally, lower baseline expression of miR-369-3p was associated with an increased likelihood of achieving pCR with Trastuzumab, while elevated expression of miR-26a-5p and miR-374-5p after two weeks of treatment correlated with pCR [84,85,86]. Zhang et al. reported that lower expression of miR-222-3p is linked to a higher likelihood of achieving pCR. When combined with data from the GeparSixto trial, the ct-miR-199a molecular profile, as outlined in the miRNA signature identified by Di Cosimo and colleagues, may enhance the ability to predict treatment responses. Therefore, reviewing these previous studies could provide valuable insights for research teams aiming to conduct future translational research on the role of miRNA signatures in predicting responses to neoadjuvant therapy [87].

This systematic review highlights the disagreement among research about the relevance of certain miRNAs, with some results directly opposing the classification of a particular miRNA as either an oncogene or a tumor suppressor. This disagreement underscores the intricate and multifaceted characteristics of miRNAs, emphasizing the necessity for researchers to focus on their varied functions in regulating biological processes [88,89,90,91]. Considering these complications, the development of a multi-miRNA panel may provide the most effective and informative strategy for enhancing the prediction of pCR after neoadjuvant therapy. Di Cosimo et al. discovered a five-miRNA signature that possesses significant, independent predictive capability for response to Trastuzumab (area under the curve [AUC]: 0.81) and Lapatinib (AUC: 0.71) in HER2+ breast cancers in the translational research component of the NeoALTTO trial. Moreover, the seminal research by McGuire et al. further illustrated the predictive capacity of several miRNAs in anticipating responses to neoadjuvant therapy across diverse breast cancer subtypes. These results collectively indicate that novel miRNA expression panels featuring clinically pertinent indicators could markedly improve the prediction of pCR in breast cancer [85].

This evidence assessment underscores the absence of consensus in executing translational research studies that assess circulating biomarker levels at different intervals during the neoadjuvant therapy period. In the GeparSixto trial, Stevic and colleagues reported findings based on pre- and post-neoadjuvant plasma ct-miRNA expression, offering insights into therapy responsiveness. Anfossi and colleagues [92] undertook tissue sample at the commencement of therapy but did not carry out further liquid or tissue biopsies during their study, whereas Jung executed interval serum sampling during the neoadjuvant, surgical, and adjuvant treatment phases in their investigation [92]. In NeoALTTO, a liquid biopsy was conducted at two weeks post-neoadjuvant therapy, yielding valuable insights into oncological outcomes and therapeutic efficacy [84]. Future multicenter translational research collaborations may benefit from the insights provided by Di Cosimo and colleagues regarding the significance of ct-miRNA as relevant biomarkers for predicting breast cancer response to neoadjuvant therapy when assessed early in the treatment process [84].

This systematic review offers a comprehensive examination of the methodological strategies employed in prior research assessing the function of miRNAs in forecasting responses to neoadjuvant treatments in HER2+ breast cancer. A principal problem in miRNA analysis is the persistent ambiguity concerning the ideal circulating media for assessment. This is evident in the variety among research, with five studies utilizing plasma, while three evaluated miRNAs in whole blood and serum. Quantitative real-time reverse transcription polymerase chain reaction (qRT-PCR) is widely acknowledged as a dependable technique for assessing miRNA expression, as demonstrated by its application in nearly all reviewed research, with a single exception. Although validated, qRT-PCR is presently regarded as suitable mainly for academic and research settings, as the absolute quantification and standardization of miRNA measurements in clinical laboratories remain goals for further progress [93]. Moreover, four of the seven prospective analyses, including the GeparSixto and NeoALTTO studies, utilized microarray methodologies to uncover non-coding miRNA targets. The utilization of microRNA microarray presents numerous advantages compared to traditional literature reviews: microarray captures relative concentration measurements of miRNA expression profiles in a designated tissue (i.e., HER2+ breast tumor tissue obtained from core biopsy), thereby enhancing the likelihood of identifying suitable molecular targets and improving the chances of achieving clinically relevant outcomes [94,95].

This comprehensive review is constrained by the inherent limitations of studies employing varied methodologies, laboratory techniques, and tissue heterogeneity. Furthermore, inconsistencies in statistical tests and analyses used in studies hinder definitive conclusions. The appropriate period for the collection of liquid or tumor biopsies remains ambiguous, since the evaluated papers yield conflicting results about the ideal tissue for extraction. Moreover, there is a persistent discussion concerning the comparative informativeness of liquid and core biopsies. The varied neoadjuvant treatment protocols and strategies utilized in the included trials limited outcome consistency, suggesting that a cautious interpretation of these findings is crucial. Despite these limitations, the authors acknowledge that this systematic analysis elucidates all previous translational research studies and forecasts miRNA responsiveness to treatment in HER2+ malignancies.

This systematic study is the inaugural comprehensive analysis of in vivo verified miRNAs capable of predicting good responses to contemporary neoadjuvant treatment regimens for patients with HER2+ breast cancer. This work emphasizes the successful aspects of prior research on miRNA’s role in the neoadjuvant context while also highlighting contentious issues that require resolution in future prospective translational studies in this field. Consequently, a comprehensive agreement delineating the role of miRNA and its potential in forecasting responses to neoadjuvant therapy in HER2+ breast cancer is still necessary. The forthcoming generation of translational research investigations may integrate both venous and core biopsy samples at specified intervals to identify stable circulating biomarkers, such as miRNAs, that could signify pCR in HER2/neu overexpressing breast cancer therapy.

### 2.6. miRNAs and Hormonal Regulation in Breast Cancer

Hormone receptor-positive breast cancer (HR+ BC) accounts for the majority of breast cancer cases and is primarily treated using endocrine therapies, including selective estrogen receptor modulators (SERMs), aromatase inhibitors (AIs), and selective estrogen receptor degraders (SERDs). Despite the proven efficacy of these therapies, endocrine resistance—both intrinsic and acquired—continues to pose a significant challenge in clinical management, contributing to recurrence and disease progression in a large subset of patients. Recent insights into the molecular mechanisms of endocrine resistance have highlighted the role of microRNAs (miRNAs) as critical modulators of estrogen receptor (ER) signaling and endocrine therapy response.

One of the most extensively studied miRNAs in this context is miR-206, which acts as a tumor suppressor and has been shown to directly target ESR1, leading to post-transcriptional downregulation of ERα. This suppression interferes with ER-mediated transcriptional activity and reduces the proliferative capacity of ER-positive cells [96,97]. Notably, high levels of miR-206 have been associated with resistance to tamoxifen and poor prognosis in HR+ breast cancer patients [98].

In addition to miR-206, the miR-221/222 cluster has emerged as a key player in endocrine resistance. These miRNAs downregulate ERα and the cell cycle inhibitor p27^Kip1, facilitating estrogen-independent growth and tamoxifen resistance [99,100]. Moreover, miR-221/222 expression is often elevated in more aggressive luminal B tumors and has been correlated with poor response to hormonal therapy in recent transcriptomic studies [101].

miR-18a, a member of the miR-17–92 cluster, has also been implicated in endocrine resistance by targeting ER coactivators and affecting chromatin remodeling at ER-binding sites [102]. Similarly, miR-22 represses ERα expression and has been proposed as a predictive biomarker for fulvestrant resistance, with recent in vitro and clinical cohort studies supporting its regulatory role in ER signaling [103,104].

More recent findings underscore the potential of circulating miRNAs as minimally invasive biomarkers of endocrine response. A 2023 multicenter study identified miR-342-3p and miR-126-5p as being associated with aromatase inhibitor resistance, suggesting their potential use in predicting therapy outcomes [105]. Another 2024 prospective trial evaluating plasma-derived exosomal miRNAs in HR+ BC patients under adjuvant letrozole therapy showed that dynamic changes in miR-503 and miR-135b levels were predictive of early relapse and could guide treatment adaptation [106].

Furthermore, advances in multi-omic analyses have identified regulatory networks involving miRNAs, long non-coding RNAs (lncRNAs), and transcription factors such as FOXA1 and GATA3, which collectively modulate ER transcriptional output and chromatin accessibility [107]. For example, miR-301a-3p was shown to promote ligand-independent ER signaling and endocrine escape via regulation of the PI3K/AKT/mTOR pathway [108].

Despite these promising findings, most miRNA-based interventions remain in preclinical stages. However, the development of miRNA mimics and antagomiRs has gained momentum, with several candidates currently undergoing early-phase clinical testing [109]. Future translational research should focus on validating miRNA panels for risk stratification and therapy monitoring, as well as exploring combinatorial strategies that integrate miRNA modulation with existing endocrine regimens.

All the mentioned microRNA information could be found in Appendix A.

## 3. Conclusions and Future Perspectives

Over time, several studies have focused on the role of different miRNAs in chemotherapy resistance or sensitivity in breast cancer. However, none of these studies have conclusively defined the exact mechanism by which these miRNAs are involved in chemosensitivity/resistance. Additionally, there is no scientific evidence regarding the existence of a correlation between molecular type and specific miRNA types that determine chemoresistance, although studies on small samples in triple-negative breast cancer and Luminal A subtype have been conducted.

Recent scientific research has identified numerous miRNAs (over 500) involved in carcinogenesis, early breast cancer detection, and treatment resistance, be it chemotherapy, hormone therapy, immunotherapy, or even radiotherapy. Small sample sizes, the lack of in vivo study frameworks, the multitude of miRNAs studied, and the lack of standardization regarding the material from which testing is performed (paraffin-embedded tissue, plasma, or whole blood) and the optimal collection times (before neoadjuvant treatment, time intervals after surgery, etc.) make data interpretation challenging in a clinical context.

Numerous miRNAs, including miR-21, miR-155, and miR-10b, function as oncogenic regulators (oncomiRs), facilitating proliferation, invasion, and immune evasion by inhibiting tumor suppressor genes such as PTEN, PDCD4, and HOXD10. These are often increased in aggressive variants, including triple-negative breast cancer (TNBC) and BRCA1-deficient tumors. In contrast, tumor-suppressive miRNAs such as those in the miR-200 family, miR-34a, and miR-503 impede critical regulators of epithelial-to-mesenchymal transition (EMT) and are frequently diminished in advanced stages of the disease. Their downregulation is associated with unfavorable prognosis and resistance to standard treatments. In hormone receptor-positive (HR+) breast cancer, miR-206, miR-221/222, miR-22, and others have been identified as modulators of estrogen receptor (ERα) signaling and as predictors of resistance to endocrine therapy, such as tamoxifen or aromatase inhibitors. Furthermore, new miRNAs like miR-301a-3p and miR-424-5p are being examined for their dual functions in immune checkpoint modulation and therapeutic resistance.

Recent studies underscore the dynamic role of microRNA (miRNA) expression patterns in predicting treatment response across different stages and subtypes of breast cancer. Evidence from both early-stage and locally advanced disease highlights the utility of baseline and treatment-modulated miRNAs as biomarkers of therapy efficacy.

In early-stage breast cancer, elevated baseline levels of miR-155 and miR-21 have been associated with stage II tumors, while miR-342 shows subtype-specific expression—being most abundant in luminal B (ER/HER2-positive) tumors and lowest in triple-negative cases [110,111]. Chemotherapy-induced reductions in miR-34a, miR-21, and miR-195 have been correlated with improved outcomes, including pathological complete response (pCR), smaller tumor sizes, and reduced proliferation [110,112].

In locally advanced disease, a combined profile of high miR-7 and low miR-340 demonstrated a 96% negative predictive value for pCR following neoadjuvant chemotherapy [113]. Additionally, the downregulation of miR-4465 during bevacizumab-based therapy has been linked to reduced cellular proliferation, suggesting a treatment-specific modulation of miRNA expression [112].

In hormone receptor-positive (HR+) breast cancer, lower levels of miR-125b-2 and miR-221 have been predictive of more favorable responses to endocrine therapy with letrozole [100]. These miRNAs may serve as clinically relevant biomarkers for tailoring hormone therapy in early and intermediate disease stages.

Overall, while inter-study heterogeneity and methodological variability limit universal conclusions, the current data support the hypothesis that miRNA signatures differ across stages and treatment types, with potential utility as predictive biomarkers. Notably, treatment-induced changes in miRNA expression—such as miR-34a or miR-155—may represent early molecular indicators of therapeutic efficacy [110,112,113]. These findings underscore the importance of further validation through standardized protocols and larger, prospective clinical studies.

Continued scientific research is needed to provide a perspective on the association of specific miRNA expressions with molecular subtypes and chemotherapy drug resistance and sensitivity related to breast cancer. This makes miRNA research clinically relevant and paves the way for new treatment strategies targeting these miRNAs and developing alternative ways to counteract chemoresistance in breast cancer (Figure 1).

## Figures and Tables

**Figure 1 ncrna-11-00078-f001:**
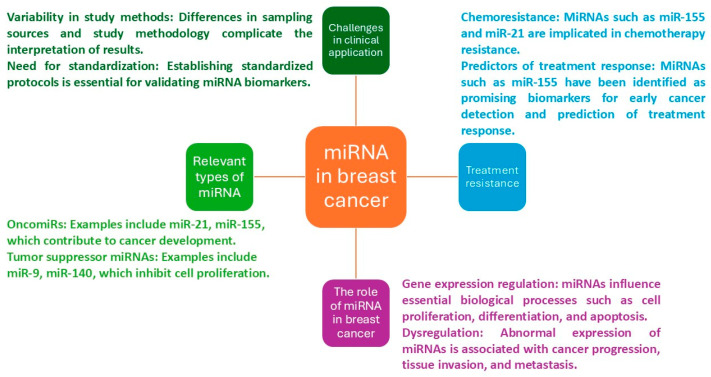
Continued research is needed to clarify the specific role of different miRNAs in molecular subtypes of breast cancer and treatment resistance.

## Data Availability

Not applicable.

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
