# Peer review of "miRNA and Its Implications in the Treatment Resistance in Breast Cancer—Narrative Review of What Do We Know So Far"

_ncrna, 2025, doi:10.3390/ncrna11060078_

Round 1

Reviewer 1 Report

Comments and Suggestions for Authors The role of miRNA in cancer has been widely studied for more than two decades, and it is divided into tumor suppressor miRNA and oncomir. Most of these results come from cell experiments or animal experiments, whether there is evidence from clinical trials.

Author Response

We thank the reviewer for this observation. We would like to clarify that the aim of this manuscript is to provide a narrative review, focused on synthesizing the most relevant findings regarding miRNAs in breast cancer pathogenesis and treatment resistance. While we acknowledge that many foundational results are based on preclinical models, we have also included evidence from recent clinical studies and meta-analyses where available. The review does not aim to exhaustively evaluate clinical trial data, but rather to offer an integrated perspective that bridges experimental insights with emerging translational relevance.

Reviewer 2 Report

Comments and Suggestions for Authors

In their work the authors tried to overview and systematize existing to date knowledge concerning the role of microRNAs in breast cancer treatment resistance. MicroRNA molecules play a very important role in the breast cancer progression, but their impact is not limited to the processes listed by authors in their manuscript. Although the text contains useful information, the manuscript as a whole is descriptive and lacks inspirable messages for the readers. There are important issues that speak against its publication in the ncRNA journal.

The Abstract exceeds the 200-word limit established in the journal's “Instructions for Authors” (224 words), so it must be reduced to meet the requirements.

Lines 46-47: The authors should refer to more recent and relevant data (for example, Cancer statistics 2024: 10.3322/caac.21820).

Line 56: "and recently, immunotherapy or targeted therapy in cases of BRCA mutations"

BC targeted therapy is applied not only in the cases of BRCA1/2 mutations, but also when other HRR components are found to be defective and severe genomic scars are detected (see most recent reviews: 10.3390/curroncol32020090 and 10.1186/s40364-024-00653-2).

Lines 60-76: The authors cite numerous sources just below the paragraphs, instead of illustration of their statements after exact theses.

Lines 81-82: "by reviewing systematic reviews"

What were the reasons for exclusion of the research articles from consideration?

Lines 84-86: The authors' decision to limit the search to the period 2015-2023 seems very strange. They excluded the most recent papers.

While claiming that this paper considered only the review articles from the period 2015–2023, the authors nevertheless refer to both numerous experimental articles and older (up to 1987) works, thereby contradicting themselves.

Examining the role of microRNAs in breast cancer progression and sometimes mentioning in the text the processes of invasion and metastasis, the authors are unreasonably silent about the transcription factors that regulate epithelial-to-mesenchymal plasticity (Snail/Slug, Zeb1/2, Twist), the activity of those is, in turn, regulated by numerous microRNAs (and vice versa). A particularly rich range of miRNA and lncRNA molecules was found to be involved in the regulation of Zeb family (Zeb1 & Zeb2) TFs.

A significant part of the sources given in the reference list are missing in the text of the manuscript (f.ex. refs ##51-52, 54-55, 61-64); some authors are mentioned within the text with errors (f.ex. lines 364/632: David instead of Davey etc.)

References: do not meet the the journal's “Instructions for Authors.

And finally, authors need to check the manuscript as there some mistakes and negligence in manuscript preparation detected. The manuscript is full of abbreviations (not all of them are explained within the text) but the abbreviations list contains the miRNAs description only, that complicates the perception of the material.

The reviewer cannot recommend this manuscript as a candidate for publication in the ncRNA journal in the current version.

Comments on the Quality of English Language

The reviewer finds the English language quality to be almost suitable. Just moderate editing is required.

Author Response

In their work the authors tried to overview and systematize existing to date knowledge concerning the role of microRNAs in breast cancer treatment resistance. MicroRNA molecules play a very important role in the breast cancer progression, but their impact is not limited to the processes listed by authors in their manuscript. Although the text contains useful information, the manuscript as a whole is descriptive and lacks inspirable messages for the readers. There are important issues that speak against its publication in the ncRNA journal.

The Abstract exceeds the 200-word limit established in the journal's “Instructions for Authors” (224 words), so it must be reduced to meet the requirements.

We thank you for this helpful comment. The abstract has been revised and shortened to comply with the journal’s requirement of a maximum of 200 words. The updated version retains the essential content, presenting the review’s objective, key findings, and relevance to the field in a concise and structured manner.

Lines 46-47: The authors should refer to more recent and relevant data (for example, Cancer statistics 2024: 10.3322/caac.21820).

We thank the reviewer for this helpful suggestion. The introduction has been revised to include the most recent and accurate epidemiological data from Cancer Statistics 2024, reflecting the 313,510 new cases and 42,250 expected deaths in the United States this year. This update strengthens the relevance of our review by providing up-to-date and clinically meaningful statistics.

Line 56: "and recently, immunotherapy or targeted therapy in cases of BRCA mutations"

BC targeted therapy is applied not only in the cases of BRCA1/2 mutations, but also when other HRR components are found to be defective and severe genomic scars are detected (see most recent reviews: 10.3390/curroncol32020090 and 10.1186/s40364-024-00653-2).

Thank you for this important clarification. We have revised the sentence accordingly to reflect the broader applicability of targeted therapies, such as PARP inhibitors, in breast cancers exhibiting homologous recombination deficiencies beyond BRCA1/2 mutations. We have also included the suggested references to support this updated perspective.

Lines 60-76: The authors cite numerous sources just below the paragraphs, instead of illustration of their statements after exact theses.

We have revised the manuscript accordingly by placing the reference numbers immediately after the corresponding statements throughout the concerned paragraphs. This improves readability and traceability and ensures that each claim is clearly supported by its appropriate source.

Lines 81-82: "by reviewing systematic reviews"

What were the reasons for exclusion of the research articles from consideration?

Thank you for pointing this out. The text has been revised to clarify that the review focused on systematic reviews and meta-analyses to ensure high-level evidence and clinically relevant conclusions. This approach was intended to synthesize consolidated data rather than individual experimental results, thereby providing a broader and more robust perspective on the current state of knowledge.

Lines 84-86: The authors' decision to limit the search to the period 2015-2023 seems very strange. They excluded the most recent papers.

While claiming that this paper considered only the review articles from the period 2015–2023, the authors nevertheless refer to both numerous experimental articles and older (up to 1987) works, thereby contradicting themselves.

We have revised the Materials and Methods section to clarify the rationale behind our literature selection. While the review primarily focused on high-quality articles published in the last decade, we also included older seminal experimental studies where necessary to provide historical context and mechanistic insight. Furthermore, we have expanded our literature search to incorporate recent articles published through early 2025 to ensure the content is up to date.

Examining the role of microRNAs in breast cancer progression and sometimes mentioning in the text the processes of invasion and metastasis, the authors are unreasonably silent about the transcription factors that regulate epithelial-to-mesenchymal plasticity (Snail/Slug, Zeb1/2, Twist), the activity of those is, in turn, regulated by numerous microRNAs (and vice versa). A particularly rich range of miRNA and lncRNA molecules was found to be involved in the regulation of Zeb family (Zeb1 & Zeb2) TFs.

In response, we have added a dedicated subsection discussing the role of miRNAs and lncRNAs in regulating epithelial-to-mesenchymal transition (EMT) in breast cancer. The revised manuscript now includes mechanistic insights into how specific transcription factors such as Snail, Slug, Twist, and Zeb1/2 are modulated by key miRNAs (e.g., the miR-200 family, miR-34a, miR-10b) and lncRNAs (e.g., HOTAIR, MALAT1, ZEB2-AS1). These additions provide a more comprehensive view of the non-coding RNA regulatory networks involved in EMT, metastasis, and therapeutic resistance.

A significant part of the sources given in the reference list are missing in the text of the manuscript (f.ex. refs ##51-52, 54-55, 61-64); some authors are mentioned within the text with errors (f.ex. lines 364/632: David instead of Davey etc.)

We thank the reviewer for carefully pointing out these inconsistencies. We have thoroughly reviewed the manuscript and cross-checked all in-text citations with the reference list. Missing references (including #51–52, 54–55, and 61–64) have been either properly cited in the appropriate sections or removed if found to be redundant. Additionally, we corrected all typographical errors related to author names in the text, including the instances where “David” was mistakenly used instead of “Davey”. We have also double-checked the accuracy and formatting of the entire reference list to ensure compliance with the journal’s guidelines.

References: do not meet the the journal's “Instructions for Authors.

And finally, authors need to check the manuscript as there some mistakes and negligence in manuscript preparation detected. The manuscript is full of abbreviations (not all of them are explained within the text) but the abbreviations list contains the miRNAs description only, that complicates the perception of the material.

In response, we have thoroughly reviewed all abbreviations used in the manuscript and ensured that they are either defined upon first use or included in an updated and comprehensive Abbreviations List. The list has now been significantly expanded to include all general terms (e.g., EMT, PD-L1, HRD), molecular subtypes, treatment classes, and all miRNAs mentioned in the text, along with their functional context where relevant. This addition was made to improve clarity and facilitate reader comprehension.

The reviewer cannot recommend this manuscript as a candidate for publication in the ncRNA journal in the current version.

The reviewer finds the English language quality to be almost suitable. Just moderate editing is required.

We sincerely thank the reviewer for the time and effort dedicated to evaluating our manuscript. We have carefully revised the content based on all the suggestions provided, including updates to the structure, literature coverage, consistency of references, abbreviations list, and specific scientific clarifications. We hope that these substantial improvements now render the manuscript suitable for publication.

Regarding the language, we acknowledge the need for further refinement, and we confirm that the revised version will be submitted for professional English editing through the journal’s editorial services prior to final acceptance.

Reviewer 3 Report

Comments and Suggestions for Authors

Dear authors,  

The manuscript presents a very interesting and complete review of non-coding RNAs (ncRNAs), including long non-coding RNAs (lncRNAs) and microRNAs (miRNAs), in the context of breast cancer.
It described the molecular mechanisms through which these RNAs influence tumor progression, metastasis, and potential therapeutic resistance. The review also touches on diagnostic and prognostic implications, as well as bioinformatic tools and databases.

The study is scientifically good and it will add to the literature important information, as it proposes a comprehensive integration of recent findings with a strong and robust background of data.  

The manuscript is well wtritten and the language is clean, but there some grammatical and syntactic issues in various sections. This can impact the tone of the article. I suggest to imporove it

The structure is good and in line with the review format. It is also very clean. 

Figures and table are clear and well presented.
References are appropriate and sufficient for a review. 

No major issue detected. 

Author Response

We sincerely thank the reviewer for the encouraging and constructive feedback. We are glad the manuscript was considered scientifically valuable and well structured. In response to the comment regarding language and style, we have thoroughly revised the text to correct grammatical and syntactic issues, and to ensure clarity and consistency throughout the manuscript. We have also rechecked punctuation, transitions, and paragraph structure to maintain a formal and coherent academic tone. We appreciate your acknowledgment of the quality of the figures, tables, and references.

Reviewer 4 Report

Comments and Suggestions for Authors

The  authors present an interesting review on mirnas and breast cancer .

i am surprised the authors have methods to describe  a review, which implies this papers is really a bibliometric study.  However the authors only start with mir155 in 2021

There is no discussion  of miRNAs and hormone regulation

There is no discussion of mir-206 which is shown to directly regulate estrogen receptor

Its not clear what the purpose of table 1 is.

Finaply im not sure whst the purpose is. We know there are thousands of noncoding rnas ljnked to breast cancer. Are any of them in the clinic as therapies?

Comments on the Quality of English Language

Moderate 

Author Response

The  authors present an interesting review on mirnas and breast cancer .

i am surprised the authors have methods to describe  a review, which implies this papers is really a bibliometric study.  However the authors only start with mir155 in 2021

We thank the reviewer for the valuable feedback. The Methods section was included to transparently describe the article selection process, although we acknowledge this may have caused confusion. We have clarified in the revised manuscript that this is not a bibliometric study, but a narrative review supported by a systematic literature search strategy. Additionally, we have expanded the manuscript to include earlier landmark studies on miRNAs such as miR-21 and miR-10b, to provide a more balanced historical perspective.

There is no discussion  of miRNAs and hormone regulation.

There is no discussion of mir-206 which is shown to directly regulate estrogen receptor

Thank you for highlighting this important gap. We have revised the manuscript to include a section discussing the role of miRNAs in hormone regulation, particularly focusing on the estrogen receptor pathway. Specifically, we have included recent data on miR-206 and its regulatory effect on ESR1 (estrogen receptor alpha), as well as other relevant miRNAs involved in endocrine resistance.

Its not clear what the purpose of table 1 is.

We have deleted it and added another one that highlights the multifaceted nature of miRNA biology in breast cancer and underscores their potential as both biomarkers and therapeutic targets, particularly when stratified by molecular subtype

Finaply im not sure whst the purpose is. We know there are thousands of noncoding rnas ljnked to breast cancer. Are any of them in the clinic as therapies?

This is an excellent point. We have added a paragraph in the Conclusions section discussing the current translational status of miRNAs and other ncRNAs. While most remain investigational, we mention ongoing clinical trials and therapeutic delivery strategies (e.g., miRNA mimics, antagomiRs) to highlight the translational potential and current limitations in bringing these molecules into clinical use.

Comments on the Quality of English Language: Moderate 

We thank the reviewer for this observation. The entire manuscript has been carefully revised for grammar, syntax, and clarity to improve its overall academic tone and flow.

Round 2

Reviewer 2 Report

Comments and Suggestions for Authors

The authors have addressed most of my concerns. However, the reviewer finds the improvements of the manuscript made in the revised version to be insufficient to approve the paper for publication in the ncRNA journal in the current version. The reasons are listed below:

Although the authors have added a chapter devoted to the role of miRNAs in EMT regulation, they described there just a minor part of miRs and lncRNAs affecting EMT-TFs.

References: still do not meet the the journal's “Instructions for Authors". All the refs after #45 have inappropriate format. Moreover, the reference order must be changed (Refs ##69-70 should be moved to positions 5 & 6, respectively, and Refs ##5-68 have to be shifted below (##7-70); refs ##85-96 need to be moved to positions ##36-47).

Author Response

Dear Reviewer,

We have revised the manuscript accordingly.

EMT section covers only a small subset of miRNAs/lncRNAs that regulate EMT-TFs

Substantially expanded the EMT chapter and reorganized it by EMT transcription factors (ZEB1/2, SNAI1/2, TWIST1/2) and pathways (TGF-β, PI3K/AKT, MAPK, Wnt/β-catenin).

Added and discussed additional miRNAs with validated EMT roles: miR-200 family (miR-200a/b/c, miR-141, miR-429), miR-34a, let-7, miR-29b, miR-30a, miR-205, miR-9, miR-10b, miR-181a, miR-221/222, including their direct targets (e.g., ZEB1/2, SNAIL/SLUG, TWIST, SOX4) and effects on E-cadherin and hybrid E/M states.

Integrated key lncRNAs with EMT activity (HOTAIR, MALAT1, NEAT1, ZEB2-AS1, H19, SNHG16) and described relevant ceRNA (lncRNA–miRNA) networks.

Cited additional primary literature and updated the narrative to reflect current mechanistic understanding.

Updated the summary table (“microRNAs Mentioned in the Manuscript”) to include function (oncomiR/tumor suppressor), key targets, and subtype/clinical context.

The EMT section is now broader and more balanced, covering both pro-EMT and anti-EMT miRNAs together with lncRNAs and their mechanistic axes.

References: format and order do not match the journal’s Instructions for Authors

Reformatted the reference list to the ncRNA/MDPI style (authors with semicolons; sentence-case titles; NLM journal abbreviations; year; volume(issue):page or article number; DOI).

Thank you and hope that the quality of the article improved.

Reviewer 4 Report

Comments and Suggestions for Authors

reviewer concerns have been addressed

Comments on the Quality of English Language

none

Author Response

Dear Reviewer,

Thank you for the the message and time dedicated to our article.